# Synthesis, Characterization, X-ray Molecular Structure, Antioxidant, Antifungal, and Allelopathic Activity of a New Isonicotinate-Derived *meso*-Tetraarylporphyrin

**DOI:** 10.3390/molecules29133163

**Published:** 2024-07-03

**Authors:** Nour Elhouda Dardouri, Soukaina Hrichi, Pol Torres, Raja Chaâbane-Banaoues, Alessandro Sorrenti, Thierry Roisnel, Ilona Turowska-Tyrk, Hamouda Babba, Joaquim Crusats, Albert Moyano, Habib Nasri

**Affiliations:** 1Laboratory of Physical Chemistry of Materials (LR01ES19), Faculty of Science of Monastir, University of Monastir, Avenue de l’Environment, Monastir 5019, Tunisia; nourelhoudadardouri@gmail.com (N.E.D.); soukaina.hrichi@gmail.com (S.H.); 2Laboratory of Medical and Molecular Parasitology-Mycology (LP3M), Faculty of Pharmacy, University of Monastir, LR12ES08, Monastir 5000, Tunisia; rajachaabanebanaoues@gmail.com (R.C.-B.); hamouda.babba@ms.tn (H.B.); 3Section of Organic Chemistry, Department of Inorganic and Organic Chemistry, Faculty of Chemistry, University of Barcelona, C. de Martí i Franquès 1-11, 08028 Barcelona, Spain; torres.yeste.pol@gmail.com (P.T.); asorrenti@ub.edu (A.S.); j.crusats@ub.edu (J.C.); 4Institute of Chemical Sciences of Rennes, UMR 6226, University of Rennes 1, Beaulieu Campus, 35042 Rennes, France; thierry.roisnel@univ-rennes.fr; 5Faculty of Chemistry, Wrocław University of Science and Technology, Wybrzeże Wyspiańskiego 27, 50-370 Wrocław, Poland; ilona.turowska-tyrk@pwr.edu.pl; 6Institute of Cosmos Science, University of Barcelona, C. de Martí i Franquès 1-11, 08028 Barcelona, Spain

**Keywords:** free base porphyrin, isonicotinic acid, X-ray molecular structure, UV visible, fluorescence, antioxidant activity, antifungal activity, allelopathic activity

## Abstract

The present article describes the synthesis of an isonicotinate-derived *meso*-arylporphyrin, that has been fully characterized by spectroscopic methods (including fluorescence spectroscopy), as well as elemental analysis and HR-MS. The structure of an *n*-hexane monosolvate has been determined by single-crystal X-ray diffraction analysis. The radical scavenging activity of this new porphyrin against the 2,2-diphenyl-1-picrylhydrazyl (DPPH) radical has been measured. Its antifungal activity against three yeast strains (*C. albicans* ATCC 90028, *C. glabrata* ATCC 64677, and *C. tropicalis* ATCC 64677) has been tested using the disk diffusion and microdilution methods. Whereas the measured antioxidant activity was low, the porphyrin showed moderate but encouraging antifungal activity. Finally, a study of its effect on the germination of lentil seeds revealed interesting allelopathic properties.

## 1. Introduction

The presence of porphyrins in myriad biological systems, coupled with the ability to finely tune their chemical and physical properties, position both free base porphyrins and metalloporphyrins as adaptable materials and as crucial for scientific investigations. The deep-rooted biological importance of these tetrapyrrolic macrocycles and their exceptional electronic and structural properties have spurred their widespread utilization, encompassing their use as photosensitizers in photodynamic therapy [1,2] and in functionalization reactions [3,4], as sensors [5,6], in solar cells [7,8], and more recently, as organocatalysts [9,10,11,12,13] and photoredox catalysts [4,14,15]. The availability of a large number of porphyrins and metalloporphyrins in these applications is made possible mainly thanks to the work of Adler and Longo [16] and Lindsey [17,18], who uncovered simple methods for the preparation of free porphyrins with acceptable yields.

In this last decade, several research groups, including our own, have undertaken a number of investigations on the use of porphyrins and metalloporphyrins as antioxidant and antifungal agents [19,20,21,22,23]. In this work, we report on the synthesis, full spectroscopic characterization, and photophysical studies of a previously unknown free base *meso*-tetraarylporphyrin: porphyrin-5,10,15,20-tetrayltetrakis(2-methoxybenzene-4,1-diyl) tetraisonicotinate, H_2_TMIPP (**1**). The molecular structure of compound **1** was determined by single-crystal X-ray diffraction of an *n*-hexane monosolvate. The antioxidant, antimicrobial, and antifungal properties, as well as the allelopathic potential, of H_2_TMIPP (**1**) have been examined.

## 2. Results and Discussion

### 2.1. Synthesis and Characterization of H_2_TMIPP (**1**)

The isonicotinate-derived free base porphyrin **1** was synthesized in two steps from commercially available vainillin (**2**) and isonicotinic acid (**3**) (Figure 1). In the first place, vainillin was directly acylated with isonicotinic acid in a dichloromethane solution, using *N*,*N*′-dicyclohexylcarbodiimide (DCC) as a coupling agent in the presence of 4-dimethylaminopyridine (DMAP) as a catalyst (10 mol%). In this way, the desired formylisonicotinate **4** was obtained in 71% yield, without the need for chromatographic purification, at the multigram scale. This procedure is somewhat more practical than the one described a few years ago by V. I. Potkin and co-workers [24], which involves the previous preparation of isonicotinoyl chloride hydrochloride. Spectroscopic data for **4** (^1^H and ^13^C NMR, HRMS-ESI) fully coincided with those in the literature [24]. Condensation of this aldehyde with freshly distilled pyrrole was performed according to the Adler and Longo method (reflux in propionic acid under open air, followed by chromatographic purification using silica gel and DCM/methanol 93:7) [16], which afforded the free base porphyrin **1** in a remarkable 29% yield.

This compound was fully characterized by IR, ^1^H-NMR, and ^13^C-NMR spectroscopy, and by HRMS-ESI spectrometry. A detailed discussion of its spectroscopic properties can be found in the Appendix A.

Recrystallization of **1** from DCM/hexane (diffusion method) afforded a single crystal that was suitable for X-ray diffraction analysis. The molecular structure of **1** showed it had crystallized with a *n*-hexane solvent molecule, a point that was also confirmed by elemental analysis.

Crystal data for C_78_H_64_N_8_O_14_, H_2_TMIPP·C_6_H_14_ (M = 1305.37 g/mol): monoclinic, space group *P*2_1_/*c* (no. 14), a = 14.9807(16) Å, b = 14.9439(16) Å, c = 14.9053(15) Å, β = 92.393(4)°, V = 3333.9(6) Å3, Z = 2, T = 150(2) K, μ(CuKα) = 0.089 mm^−1^, Dcalc = 1.300 g/cm^3^, and 22,628 reflections measured (1.926° ≤ 2Θ ≤ 25.000°), of which 5858 were unique (R_int_ = 0.0596) and used in all calculations. The final R_1_ was 0.1102 (I > 2σ(I)) and wR_2_ was 0.2811 (all data) [25].

A detailed discussion of the molecular structure of the hexane monosolvate of **1** can be found in the Appendix A.

### 2.2. Photophysical Properties of H_2_TMIPP (**1**)

#### 2.2.1. UV/Vis Spectroscopy

The electronic spectrum of the H_2_TMIPP (**1**) free base porphyrin in DCM solution is represented in Figure 1, and the UV/Vis data of this *meso*-tetraarylporphyrin, as well as those reported for structurally related compounds, are summarized in Table 1. The λ_max_ value of the Soret band of compound **1** is 420 nm and those of the Q bands are 516, 551, 591, and 647 nm, respectively. These values are very close to those of the known *meso*-tetraarylporphyrins (Table 1) [19,26,27,28,29], which is an indication that all *meso*-tetraarylporphyrins exhibit very similar UV/Vis spectra irrespective of the nature of the substituents in the *ortho*-, *meta*-, or *para*-positions of the phenyl rings in these systems.

On the other hand, we determined the optical gap energy (Eg-op) of H_2_TMIPP (**1**) by means of the Tauc plot method [30]. The measured value, 1.881 eV, is typical for a free base *meso*-tetraarylporphyrin.

#### 2.2.2. Fluorescence Spectroscopy

The emission spectrum of compound **1** is depicted in Figure 2, while Table 2 summarizes the emission band maxima (λ_max_), fluorescence quantum yields (ϕ_f_), and lifetime (τ_f_) values of our free base porphyrin (**1**), together with those corresponding to a selection of *meso*-tetraarylporphyrins.

An inspection of Table 2 shows that for all *meso*-arylporphyrins reported in this Table, including our synthetic free base porphyrin **1**, the λ_max_ values of the Q(0,0) band are around 655 nm while the λ_max_ values of the Q(0,1) band are close to 716 nm. This is an indication that the positions of the Q(0,0) and Q(0,1) emission bands are independent of the type of the substituent in the *para* positions of a *meso*-arylporphyrin. Concerning the fluorescence quantum yields (ϕ_f_), the value of the unsubstituted *meso*-tetraphenylporphyrin (H_2_TPP) is the highest (ϕ_f_ = 0.11), while the substituted *meso*-tetrarylporphyrins exhibit smaller values of the fluorescence quantum yields. This could be explained by the quenching effect of *para*-phenyl substituents of *meso*-arylporphyrin.

### 2.3. Biological Properties of H_2_TMIPP (**1**)

#### 2.3.1. Antioxidant Activity

To evaluate the antioxidant properties of the free base porphyrin H_2_TMIPP (**1**), we used the 1,1-diphenyl-2-picrylhydrazyl (DPPH) radical assay. The stable DPPH nitrogen-centered radical exhibits a characteristic absorption pattern that diminishes in the presence of antioxidants [31]. By reacting with hydrogen-donating functionalities, the color of the DPPH changes from purple to yellow, and the intensity of the color change is related to the number of electrons captured [32].

The radical scavenging activity of H_2_TMIPP (**1**) and ascorbic acid against DPPH as a function of the concentration is described in Figure 3, which shows that the rate of free radical trapping increases with the concentration of the inclusion compound solution. We notice that the porphyrin scavenging performance on DPPH radicals increased very weakly from 27.89% (at 2 mg/mL) to 22.05% (at 1 mg/mL) and 19.07% (at 0.5 mg/mL), with a half-maximal inhibitory concentration (IC_50_) value of 3.6412 ± 0.4256 mg/mL, compared to 0.01321 ± 0.0062 mg/mL using ascorbic acid (reference). The fact that the H_2_TMIPP free base porphyrin exhibits weak antioxidant properties could be explained by the presence of carbonyl groups and the large size of this porphyrin. Indeed, it has been observed that the presence of electron-donating substituents in the phenyl groups of a *meso*-arylporphyrin increases its antioxidant activity, while electron-withdrawing substituents decrease it [33]. Our synthetic porphyrin presents methoxy groups (MeO) which are electron donors and carboxyl groups which have a strong affinity of electrons. Therefore, these carbonyl moieties will attract electrons from the methoxy groups and the H_2_TMIPP free base porphyrin will behave as an electron-withdrawing species, leading to a decrease in the antioxidant activity of our synthetic porphyrin.

#### 2.3.2. Antifungal Activity

##### The Disk Diffusion Method

The antifungal activities of the H_2_TMIPP (**1**) free base porphyrin were quantitatively assessed by the presence or absence of zones of inhibition against three yeast strains (*Candida albicans* ATCC 90028, *Candida glabrata* ATCC 64677, and *Candida tropicalis* ATCC 64677). The diameters of the inhibition zones observed by the disk diffusion method are shown in Table 3. The solvent used to prepare the porphyrin solution is dimethyl sulfoxide (DMSO), which showed no inhibition against the test organisms (negative control); Amphotericin B, a commercially available antifungal antibiotic, was used as a reference. The data presented in Table 3 show that H_2_TMIPP **1** gives rise to almost the same pronounced inhibition zones against *C. albicans* and *C. tropicalis*, where the inhibition zone is 12 ± 0.5 and 12 ± 0.4 mm, respectively, which are different from the effect of the reference drug (Amphotericin B), with inhibition zones of 20 ± 1 and 20 ± 0.6 mm for the same fungal strains *C. albicans* and *C. tropicalis*, respectively. However, it is less active against *C. glabrata*, with an inhibition zone of 8 ± 0.3 mm. The disk method enabled us to determine whether compound **1** had fungal potential, functioning as a screening for its antifungal activity. Indeed, this porphyrin showed moderate antifungal activity against all three *Candida* species. This effect is an indication of the compound’s ability to inhibit fungal growth. 

##### The Microdilution Method

The study carried out on the antifungal activity of the free porphyrin H_2_TMIPP (**1**) and Amphotericin B (used as a reference), using the microdilution method, revealed significant results on three yeast strains of the *Candida* genus: *Candida albicans*, *Candida glabrata* and *Candida tropicalis*. For compound **1**, the minimum inhibitory concentrations (MICs) and the minimum fungicidal concentrations (MFCs) varied depending on the strain. *Candida albicans* and *Candida tropicalis* showed greater sensitivity, with a MIC of 1.25 mg/L and a MFC of 2.5 mg/L, while *Candida glabrata* showed greater resistance, with a MIC of 5 mg/L and a MFC greater than 5 mg/L. These results suggest a variable efficacy of our isonicotinate-derived porphyrin depending on the strain (Table 4). For Amphotericin B, the results indicate satisfactory antifungal activity, although slightly higher concentrations are required for *Candida glabrata*. These significant antifungal activities observed for our compound **1** may be due to the presence of the NH group in the porphyrin core of H_2_TMIPP, which plays an important role in the antimicrobial activity. In addition, this *meso*-tetraarylporphyrin skeleton also contains the phenyl isonicotinate moiety, which may also be responsible for the higher antifungal activity of this porphyrin. 

#### 2.3.3. Allelopathic Activity

Figure 4 shows the effects of different concentrations of the free base porphyrin H_2_TMIPP (**1**) on the germination, above-ground growth, root growth, and hydration of lentil seeds. Firstly, regarding the germination (%G) (Figure 4a), no inhibition is observed at a 0.625 mg/mL concentration, but from the 1.25 mg/mL concentration onwards, a progressive inhibition is observed, reaching 50% at a concentration of 10 mg/mL. This indicates that H_2_TMIPP has a significant impact on lentil seed germination. Concerning the growth of the aerial part, even at a concentration of 0.625 mg/mL, a significant inhibition of 37.94% was observed, and this inhibition increases linearly with the increase in the concentration of H_2_TMIPP. At a concentration of 10 mg/mL, inhibition reached 71.7% (Figure 4c), underlining the major impact on the growth of the aerial part of the seedlings. Similarly, root growth (Figure 4b) showed an increase in inhibition as the concentration of free base porphyrin **1** increased. At a concentration of 10 mg/mL, inhibition reached 74.13%, highlighting a significant impact on seedling root development. In terms of the hydration (%H) of the seedlings (Figure 4d), the percentage of hydration ranged from 66.87% to 71.37%. This indicates that porphyrin **1** negatively affects the hydration capacity of lentil seedlings.

These results clearly reveal the allelopathic potential of H_2_TMIPP, suggesting that it acts as a growth-inhibiting agent in lentil seeds. As porphyrins are naturally present in plants chlorophyll, these results raise important questions about their potential role as allelopathic agents in nature. It is possible that porphyrins play a role in inter-species competition by influencing the growth of neighboring plants, which could have implications for understanding the mechanisms of interaction between plants within ecosystems.

## 3. Materials and Methods

### 3.1. General Methods

Commercially available reagents, catalysts, and solvents were used as received from the supplier. Dichloromethane for porphyrin synthesis was distilled from CaH_2_ prior to use, and THF was dried by distillation from LiAlH_4_. Deuterated solvents were supplied by Merck Life Science (Mollet del Vallés, Spain).

Thin-layer chromatography was carried out on silica gel plates Merck 60 F_254_, and compounds were visualized by irradiation with UV light and/or chemical developers (KMnO_4_, *p*-anisaldehyde, and phosphomolybdic acid). Chromatographic purifications were performed under pressurized air in a column with silica gel Merck 60 (particle size: 0.040–0.063 mm, Merck Life Science S.L., Mollet del Vallés, Spain) as stationary phase and solvent mixtures (hexane, ethyl acetate, dichloromethane, and methanol) as eluents.

^1^H (400 MHz) NMR spectra were recorded with a Varian Mercury 400 spectrometer (Agilent Technologies, Santa Clara, CA, USA). Chemical shifts (δ) are provided in ppm relative to the peak of tetramethylsilane (δ = 0.00 ppm), and coupling constants (*J*) are provided in Hz. The spectra were recorded at room temperature. Data are reported as follows: s, singlet; d, doublet; t, triplet; q, quartet; m, multiplet; and br, broad signal. IR spectra were obtained with a Nicolet 6700 FTIR instrument (Thermo Fisher Scientific, Waltham, MA, USA), using ATR techniques. UV–vis spectra were recorded on a double-beam Cary 500-scan spectrophotometer (Varian, Palo Alto, CA, USA). Cuvettes (quartz QS Suprasil, Hellma, Hellma GmbH & Co., KG, Mülheim, Germany) cm were used for measuring the absorption spectra. The porphyrin solutions in water were carefully degassed by gently bubbling a nitrogen gas stream prior to the spectrophotometric measurement.

### 3.2. Synthetic Procedures and Product Characterization

#### 3.2.1. Synthesis of 4-Formyl-2-methoxyphenyl Isonicotinate **4**

Isonicotinic acid **3** (6.03 g, 0.049 mol), 4-hydroxy-3-methoxybenzaldehyde **2** (7.45 g, 0.049 mol), and 4-dimethylaminopyridine (DMAP) (0.6g, 0.0049 mol) were dissolved in dry dichloromethane (40 mL) at 0 °C. To this solution, a solution of *N*,*N*′-dicyclohexylcarbodiimide (DCC) (10.11 g, 0.049 mol) in dry dichloromethane (25 mL) was added dropwise; when the addition was finished, the mixture was stirred at room temperature for 12 h. Upon completion of the reaction, the resulting mixture was filtered, and the dichloromethane was removed by rotary evaporation. The residue was poured over water, and the solid was collected by filtration, washed with water, followed by *n*-hexane, and dried under vacuum to afford a pale-yellow powder (9.0 g, yield 71.4%), whose spectral data coincided with those described in the literature [24]. 

^1^H NMR (CDCl_3_, 400 MHz): *δ* = 10.0 (s, 1H), 8.88 (dd, *J* = 4.4, 1.7 Hz, 2H), 8.01 (dd, *J* = 4.4, 1.6 Hz, 2H), 7.56 (dd, *J* = 3.4, 1.7 Hz, 1H), 7.54 (d, *J* = 1.8 Hz, 1H), 7.36 (d, *J* = 7.9 Hz, 1H) 3.90 (s, 3H) ppm. ^13^C NMR (CDCl_3_, 151 MHz): *δ* = 190.89 (C=O of CHO), 162.73 (C=O of ester), 151.89, 150.86(x2), 144.51, 138.26, 136.08, 135.62, 124.68, 123.31(x2), 110.98, 56.14 (C–O of methoxy) ppm. HRMS [ESI^+^]: *m*/*z* calcd for C_14_H_12_NO_4_ [M + H]^+^: 258.0761 found: 258.0769.

#### 3.2.2. Synthesis of Porphyrin-5,10,15,20-tetrayltetrakis(2-methoxybenzene-4,1-diyl) Tetraisonicotinate **1**

H_2_TMIPP (**1**) was prepared by means of the Adler and Longo method [16]. In a 250 mL three-necked flask, 4-formyl-2-methoxyphenyl isonicotinate **4** (3.72 g, 14.4 mmol) was dissolved in propionic acid (50 mL), and the open-air solution was heated under reflux at 140 °C. Freshly distilled pyrrole (1.0 mL, 14.5 mmol) was then added dropwise, and the resulting mixture was stirred under reflux for another 40 min. The mixture was cooled down at room temperature. Propionic acid was distilled under vacuum and the obtained solid was dissolved in 20 mL of dichloromethane and then neutralized with a 1 M aqueous solution of ammonia. After decantation, the organic phase was concentrated, and the residue was purified by column chromatography on silica gel (dichloromethane /methanol 93/7 as eluent). A purple solid was obtained and dried under vacuum overnight (1.26 g, 29% yield). The single crystal X-ray molecular structure of compound **1**, recrystallized from hexane/DCM, shows that it co-crystallizes with an *n*-hexane solvent molecule.

FTIR (solid, $\bar{\nu \;}({cm}^{-1})$ = 3316 ν(NH) (porphyrin), 2960 ν(CH) (porphyrin), 1736 ν(C=O) (ester), 1263–1063, ν(C–O) (ester), 967 δ(CCH) (porphyrin). ^1^H NMR (CDCl_3_, 400 MHz) δ (ppm): 9.0 (s, 8H, H-pyrrole), 8.97 (dd, *J*_1_ = 4.4 Hz, *J*_2_ = 1.6 Hz, 8H Ar), 8.21 (dd, *J*_1_ = 4.4Hz, *J*_2_ = 1.6 Hz, 8H Ar), 7.92 (s, 4H Ar), 7.89 (d, *J* = 8.0 Hz, 4H Ar), 7.58 (d, *J* = 8.0 Hz, 4H Ar), 3.95 (s, 12H, 4H-OCH_3_), −2.77 (s, 2H, NH). ^13^C NMR (151 MHz, CDCl_3_) δ (ppm): 163.51, 150.87(x2), 151.5, 149.2, 141.22, 139.5, 136.77, 127.21, 123.51(x2), 121.0, 120.5, 119.3, 119.2, 56.23 ppm. HRMS [ESI^+^]: *m*/*z* calcd for C_72_H_51_N_8_O_12_ [M + H]^+^: 1219.3621 found: 1219.3617 (−0.39 ppm). UV/Vis [λ_max_ (nm) in CH_2_Cl_2_, (log ε)]: 420 (5.73), 516 (4.33), 551 (3.97), 591 (3.87), 647 (3.68). Elemental analysis calcd (%) for C_78_H_64_N_8_O_16_ [H_2_TMIPP^•^C_6_H_14_]: C 71.77, H 4.79, N 8.58; found: C 72.02, H 4.93, N 8.89.

### 3.3. Antioxidant Activity

The 1-diphenyl-2-picrylhydrazyl (DPPH) scavenging activity was determined according to the method described by Hrichi et al. [34], with minor modifications. Both for synthetic H_2_TMIPP (**1**) free base porphyrin and for ascorbic acid (used as a standard), a series of dilutions were performed in ethanol to obtain 10 different concentrations, ranging from 2.00 to 0.0039 mg/mL^−1^. One hundred microliters of a 0.1 mM solution of DPPH in ethanol was added to 100 µL of each concentration placed in a 96-well microplate. The mixture was shaken gently and incubated in the dark for 30 min. The absorbance of the reaction mixture was measured at 517 nm using a microplate spectrophotometer (Multiscan FC, Thermo-Scientific). Ethanol was used as a reagent blank in place of the sample, the control being a DPPH/ethanol (v/v) mixture. Trials were carried out in triplicate. 

The percentage of free radical scavenging by the test samples was calculated using the Equation (1): (1)$$DPPH\;radical\;scavenging\;\%=\frac{\left(A_{control}-A_{sample}\right)}{A_{control}}\times 100$$
where *A_control_* represents the absorbance of the control sample and *A_sample_* represents the absorbance of the sample under test. The IC_50_ (half-maximal inhibitory concentration) value was determined as the concentration at which each sample exhibits 50% radical scavenging activity.

### 3.4. Antifungal Activity


**
*Fungal strain and culture conditions:*
**


The in vitro antifungal activities of H_2_TMIPP (1) and Amphotericin B, a commercially available antifungal antibiotic used as a reference, were tested against three human yeast strains belonging to the *Candida* (C.) genus, obtained from the American Type Culture Collection, specifically *C. albicans* (ATCC 90028), *C. glabrata* (ATCC 64677), and *C. tropicalis* (ATCC 66029). Before conducting each antifungal test, these fungal strains were subcultured on Sabouraud dextrose Chloramphenicol plates (Biolife). 


**
*Disk Diffusion Method:*
**


A two hundred micro-liter suspension of *Candida* spp. containing 0.5 McFarland was uniformly dispersed on the MH (Muller Hinton Agar) medium. A six-millimeter diameter paper disc was impregnated with 10 µL of 100 µg/mL porphyrin solutions, dried, and placed on the MH medium previously inoculated with the *Candida* spp. culture. The medium was incubated at 37 °C for 24 h and the inhibition activity was measured using the mean diameter of the inhibition zone. Standard Amphotericin B disc was used as a positive control for antifungal activity.


**
*Determination of Minimum Inhibitory Concentration (MIC) and Minimum Fungicidal Concentration (MFC)*
**


The microdilution technique was used to determine the minimum inhibitory concentrations (MICs) of the free base porphyrin on *Candida* spp. strains according to a modified Hrichi et al. [35] method. Firstly, solutions of H_2_TMIPP dissolved in dimethyl sulfoxide (DMSO) (*w*/*v*) were prepared and then diluted in RPMI 1640 supplemented with 2% glucose. Then, 100 μL of each porphyrinic solution was dispensed to each well at concentrations between 5 and 0.039 mg·mL^−1^. Next, 90 μL of suspension adjusted to 1-2.5 x 105 CFU mL^−1^ in RPMI 1640-2% glucose was added to each well. Finally, 10 μL of resazurin indicator solution (0.37 g·mL^−1^) was added to reach a final volume of 200 μL per well. A negative control was obtained by adding 200 μL of RPMI 1640-2% glucose to a well and a positive control by adding 100 μL of RPMI 1640-2% glucose and 100 μL of inoculum. The microplates were incubated at 37 °C for 24 h. For each strain tested, three replicates were performed. The MIC was considered to be the lowest concentration of porphyrinic solution at which each candida spp did not grow. To determine the FMC, 10 μL of each concentration after the concentration corresponding to the MIC was subcultured onto Sabouraud Dextrose Agar (SDA) plates and incubated at 37 °C for 72 h. As a result, the MFC was defined as the H_2_TMIPP concentration that gave rise to no visible colonies or revealed three or fewer colonies, providing approximately 99–99.5% growth inhibition.

### 3.5. Allelopathic Activity of H_2_TIMPP

The allelopathic activity of the H_2_TIMPP free base porphyrin was tested in vitro, using seeds of the model plant lens culinaris Medik (lentils), according to the method described by Hrichi et al. [34]. Five concentrations (5, 2.5, 1.25, 0.625, and 0.3125 mg·mL^−1^) of each porphyrin dissolved in methanol were prepared and added to 9.0 cm diameter Petri dishes. Control tests were carried out using distilled water. After evaporation of the solvent, 10 seeds and 5 mL of distilled water were placed on the filter paper disc, thus maintaining the initial concentration. All tests were carried out in triplicate. The Petri dishes were left in ambient temperature, light, and humidity conditions in the laboratory. The data relating to seed germination, shoot length, root length, and hydration were recorded 7 days after sowing. We counted the number of germinated seeds and then determined the germination percentage using Equation (2):(2)$$\%G=\left[\frac{NGS}{TNS}\times 100\right]$$
where NGS is the number of germinated seeds and TNS is the total number of seeds.

Germination inhibition percentages were calculated using Equation (3):(3)%I=100−%G

The plants were carefully sampled, and the lengths of the radicles and hypocotyls were measured with a ruler and expressed in centimeters (cm). Percent elongation was determined using Equation (4):(4)$$\%I=\left[1-\frac{\left(E\right)}{T}\times 100\right]$$
where *E* is the value of the parameter studied (length of aerial part, length of root part) in the presence of the extract and *T* is the value of the parameter studied (length of aerial part, length of root part) in the presence of the control (distilled water). 

The fresh mass was determined by weighing the plants on an analytical balance. The plants were then dried in a forced-air oven at 60 °C for 24 h and weighed again to obtain the dry mass.

The percentage hydration was calculated using Equation (5):(5)$$\%H=\left[\frac{\left(PF-PS\right)}{PF}\right]\times 100$$
where *PF* is the fresh sample weight and *PS* is the dry sample weight. The percentage of hydration inhibition was calculated according to Equation (6):(6)%I=100−%H

For the %*I* > 0, there is inhibition, and for the %*I* < 0, there is stimulation.

### 3.6. X-ray Diffraction

A dark blue prism-shaped single crystal of compound **1** (co-crystal with *n*-hexane) was used for an X-ray diffraction investigation. The data were collected at 150(2) K on a D8 VENTURE Bruker AXS diffractometer using Mo Kα radiation of wavelength 0.71073 Å. The SADABS program (Bruker AXS 2014, Bruker, Karlsruhe, Germany) [36] was used for the absorption correction. The SIR-2014 program was used to solve the structure of compound **1** [37] and the SHELXL-2014 program [38] was used to refine this structure by full-matrix least-squares techniques on F^2^. During the refinements, we noticed that one arm of the porphyrin made by a phenyl, an ester, a methoxy, and a pyridyl group is disordered in two positions [(C31A-C32A-C33A-C24A-C35A-C36A-O37A-C38A-O39A-C40A-O41A-C42A-C43A-C44A-N45A-C46A-C47A) and (C31B-C32B-C33B-C24B-C35B-C36B-O37B-C38B-O39B-C40B-O41B-C42B-C43B-C44B-N45B-C46B-C47B)] with refined position occupancies of 0.590 (6) and 0.410 (6), respectively. Furthermore, the *n*-hexane molecule is disordered over three positions [C48A-C49A-C50A), (C48B-C49B-C50B), and (C51-C52-C53)] with refined position occupancies of 0.236 (3), 0.265 (3) and 0.499 (3), respectively. For these two disordered moieties, the anisotropic displacement ellipsoids of the disordered atoms are very elongated, which indicates that they are statistically disordered. Consequently, for the disordered phenyl, the SIMU and PLAT restraints commands in the SHELXL-2014 software were used [39]. The DFIX and DANG constraint commands were also used to correct the geometry of these disordered fragments [39].

For compound **1**, non-hydrogen atoms were refined with anisotropic thermal parameters, whereas H-atoms were included at estimated positions using a riding model. Drawings were made using ORTEP3 for Windows [40] and MERCURY [41].

## 4. Conclusions

In conclusion, we have successfully prepared a new *meso*-tetraarylporphyrin, namely the porphyrin-5,10,15,20-tetrayltetrakis(2-methoxybenzene-4,1-diyl) tetraisonicotinate H_2_TMIPP (**1**). The spectroscopic properties of this free base porphyrin were studied by ^1^H and ^13^C NMR, IR, UV/Vis, and fluorescence. The structure of this porphyrinic species was elicited by high-resolution ESI Mass spectrometry and by single crystal X-ray diffraction. DPPH was used to test the scavenging activity of H_2_TMIPP against three yeast strains (*C. albicans* ATCC 90028, *C. glabrata* ATCC 64677, and *C. tropicalis* ATCC 64677). This compound shows rather weak antioxidant properties compared to ascorbic acid, used as a reference. This free base porphyrin (using both the disk diffusion and the microdilution methods) showed moderate but encouraging antifungal activity against the same three yeast strains, compared to that of Amphotericin B which was used as a reference. The allelopathic properties of H_2_TMIPP were also studied on the germination, above-ground growth, root growth, and hydration of lentil seeds. This investigation led to the following results: (i) concerning the germination, at a H_2_TIMM concentration of 10 mg/mL, we obtained an inhibition percentage of 50%; (ii) with regard to the growth of the aerial part, for a concentration of 10 mg/mL of H_2_TMIM, the inhibition reached 71.7%; (iii) for the root growth of lentil seeds, for the same 10 mg/mL concentration of H_2_TMIPP, inhibition reached 74.13%; and (iv) in terms of the hydration (%H) of the seedlings, the hydration percentage is between 66.87 and 74.13%. Therefore, this new free base *meso*-tetraarylporphyrin exhibits very interesting allelopathic properties on lentil seeds.

## Data Availability

No new data were created in addition to those reported here and in the Appendix A.

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
