# Peer review of "Synthesis, Characterization, X-ray Molecular Structure, Antioxidant, Antifungal, and Allelopathic Activity of a New Isonicotinate-Derived meso-Tetraarylporphyrin"

_molecules, 2024, doi:10.3390/molecules29133163_

Round 1
Reviewer 1 Report
Comments and Suggestions for Authors
I recognize the importance of the results and their contribution to the research field. I commend the authors for their careful presentation of the results and the extensive criteria used for characterization.
1a) I did not detect any typographical errors, except DDPH instead of DPPH. Abstract, line 28; Results and discussion, line 160; and Conclusions, line 44.
2b) My main concern is why the researchers compared the results of the test compound with those of the reference compound in different concentration units (Table 4). Line: 331 - I suggest that the concentrations be expressed in micromolar so that a proper comparison can be made.
3c) Conclusions: Since the authors' research group has been working with porphyrin derivatives, evaluating their antifungal and antioxidant activities, it would be interesting to compare the results described in this manuscript with previous ones. I did not detect this discussion. Just a suggestion.
Author Response
1a) I did not detect any typographical errors, except DDPH instead of DPPH. Abstract, line 28; Results and discussion, line 160; and Conclusions, line 44.
These typos have been corrected in the Manuscript.
2b) My main concern is why the researchers compared the results of the test compound with those of the reference compound in different concentration units (Table 4). Line: 331 - I suggest that the concentrations be expressed in micromolar so that a proper comparison can be made.
All concentrations in Table 4 have now been expressed in micrograms/L.
3c) Conclusions: Since the authors' research group has been working with porphyrin derivatives, evaluating their antifungal and antioxidant activities, it would be interesting to compare the results described in this manuscript with previous ones.Unfortunately, it is not possible to compare the antimicrobial activity of H2TMIPP with those of reported free base porphyrins. This is because our free base porphyrin was tested on Candida albicans, Candida glabrata and Candida tropicalis yeast strains which is not the case of the reported investigations using other free base porphyrins for which the antimicrobial activity where tested using other strains.
Reviewer 2 Report
Comments and Suggestions for Authors
The manuscript provides thorough characterization of the synthesized porphyrin using various spectroscopic techniques and single-crystal X-ray diffraction, confirming its identity and purity. It evaluates multiple biological activities, demonstrating potential multifunctional applications, with notable antifungal and allelopathic properties. The synthesis method is well-documented, ensuring reproducibility, with the yield and purity of the final product adequately reported.
But it also has the following weaknesses:
· Antioxidant Activity: The manuscript notes low antioxidant activity, which limits potential applications. The evaluation is interesting, but including additional controls or reference compounds would put the observed effects into better context.
· Limited Antifungal Scope: The study reports antifungal activity against three yeast strains. Including a wider range of fungal species would better assess the compound’s antifungal spectrum. Including additional controls or reference compounds would put the observed effects into better context.
· Allelopathic Activity Assessment: The allelopathic study on lentil seeds is intriguing but limited. Including additional plant species and a more detailed analysis would add depth to the findings.
The following improvements can make the report more robust:
1. Conduct mechanistic studies or molecular docking simulations to better understand the mode of action and potential targets for the observed antifungal and allelopathic activities.
2. Evaluate the compound's photophysical properties, such as singlet oxygen generation and photodynamic activity, which could be valuable given the potential applications of porphyrins in this area
Comments on the Quality of English Language
Mostly ok.
Author Response
The following improvements can make the report more robust:1. Conduct mechanistic studies or molecular docking simulations to better understand the mode of action and potential targets for the observed antifungal and allelopathic activities.
2. Evaluate the compound's photophysical properties, such as singlet oxygen generation and photodynamic activity, which could be valuable given the potential applications of porphyrins in this area.
Answer: We completely agree with the referee that these suggested additional experiments would make the manuscript more comprehensive, and we are grateful for these suggestions about further developments of our research. However, we cannot deal with these issues within the framework of minor revisions of the present manuscript, since we would have to launch a new research program.